Brain transcriptomes of harbor seals demonstrate gene expression patterns of animals undergoing a metabolic disease and a viral infection

Rosales Stephanie M. srosales712@gmail.com
Vega Thurber Rebecca L.
Department of Microbiology, Oregon State University , Corvallis , OR , United States
Prentis Peter
Electronic publication date: 2016 Dec 22
Publication date: 2016
Volume: 4
Electronic Location ID: e2819
Received 2016 Jul 8; Accepted 2016 Nov 22
Copyright: ©2016 Rosales and Vega Thurber
Copyright year: 2016
Copyright holder: Rosales and Vega Thurber
License: This is an open access article distributed under the terms of the Creative Commons Attribution License, which permits unrestricted use, distribution, reproduction and adaptation in any medium and for any purpose provided that it is properly attributed. For attribution, the original author(s), title, publication source (PeerJ) and either DOI or URL of the article must be cited.
License URL: https://creativecommons.org/licenses/by/4.0/

Keywords: Marine mammal, Phocine herpesvirus, Fatty acid metabolism, High-throughput sequencing, Burkholderia

Funding: Sea Grant Starter NA010AR4170059 NSF Graduate Research Fellowship 2012136295 This work was supported by Sea Grant Starter Grant # NA010AR4170059 NA223B R700 to RVT and an NSF Graduate Research Fellowship #2012136295 to SMR. The funders had no role in study design, data collection and analysis, decision to publish, or preparation of the manuscript.

==============================
Diseases of marine mammals can be difficult to diagnose because of their life history and protected status. Stranded marine mammals have been a particularly useful resource to discover and comprehend the diseases that plague these top predators. Additionally, advancements in high-throughput sequencing (HTS) has contributed to the discovery of novel pathogens in marine mammals. In this study, we use a combination of HTS and stranded harbor seals (Phoca vitulina) to better understand a known and unknown brain disease. To do this, we used transcriptomics to evaluate brain tissues from seven neonatal harbor seals that expired from an unknown cause of death (UCD) and compared them to four neonatal harbor seals that had confirmed phocine herpesvirus (PhV-1) infections in the brain. Comparing the two disease states we found that UCD animals showed a significant abundance of fatty acid metabolic transcripts in their brain tissue, thus we speculate that a fatty acid metabolic dysregulation contributed to the death of these animals. Furthermore, we were able to describe the response of four young harbor seals with PhV-1 infections in the brain. PhV-1 infected animals showed a significant ability to mount an innate and adaptive immune response, especially to combat viral infections. Our data also suggests that PhV-1 can hijack host pathways for DNA packaging and exocytosis. This is the first study to use transcriptomics in marine mammals to understand host and viral interactions and assess the death of stranded marine mammals with an unknown disease. Furthermore, we show the value of applying transcriptomics on stranded marine mammals for disease characterization.

Introduction

The combination of high-throughput sequencing (HTS) and stranded marine mammals for disease discovery

The health of wild marine mammal populations is difficult to assess because of their unknown population sizes, large distributions, and protected status. Stranded or vulnerable animals found ashore, have been essential for scientists to identify causes of marine mammal deaths. For example, important pathogens like phocine distemper virus (PDV), and Leptospira, were originally discovered in stranded marine mammals (Vedros et al., 1971; Osterhaus et al., 1988). Yet, a large majority of marine mammal deaths remain unknown. In 2007, it was reported that only 56% of marine mammal mortality events had a known cause of death (Gulland & Hall, 2007), leaving the pathogens and physiological causes of many diseases to be discovered. However, the introduction of high-throughput sequencing (HTS) has led to the identification of many more marine mammal pathogens, such as seal and California sea lion anellovirus, phocine herpesvirus 7, and seal parvovirus (Ng et al., 2009; Ng et al., 2011; Bodewes et al., 2013; Kuiken et al., 2015). Therefore, the combination of stranded animals and HTS are vital resources for the discovery of marine mammal diseases.

HTS and gene expression studies to understand disease in marine mammals

Although the discovery of new diseases can aid in the conservation of marine mammal populations, there is also a need to further describe known marine mammal diseases that are not fully understood. For example, phocine herpesvirus-1 (PhV-1) was discovered in 1985 and is highly abundant in North American harbor seal adults (99%). It is particularly pathogenic to young seals causing ∼46% mortality (Osterhaus et al., 1985; Harder et al., 1996; Gulland et al., 1997). Despite PhV-1′s deleterious impacts, we have little understanding of the effects of PhV-1 on host gene expression. Previous marine mammal studies have identified pinniped immune responses against PhV-1 using enzyme-linked immunosorbent (ELISA), but this offers minimal information about the disease (Harder et al., 1998). Marine mammal studies have applied targeted gene expression techniques using RT-qPCR to examine immune and endocrine responses to immunotoxins and physiological changes (Neale et al., 2005; Hammond, 2005; Tabuchi et al., 2006). However, ELISA and RT-qPCR target a limited number of host gene expressions; thus they do not represent the global host response.

HTS is a powerful resource for assessment of both the etiology of a disease and the response of the host during disease events. For example, using transcriptomic analysis, scientists were able to determine that a toxin caused a mass mortality event of abalone, pinpoint the origin of the toxin, and access the genetic effects on the abalone population (De Wit et al., 2014). However, transcriptomic analysis has rarely been used to comprehend the effects of stressors and diseases on marine mammal health. There has been some increase in marine mammal transcriptomic studies, such as a study by Hoffman et al. (2013) which suggested that post-mortem samples can be reliable resources for genomic studies. Yet, there have been few studied that use transcriptomics to measure physiological stress responses in marine mammals, and there have been no studies that looked at pathogen responses in these megafauna (Mancia et al., 2014; Niimi et al., 2014; Khudyakov et al., 2015a; Khudyakov et al., 2015b; Fabrizius et al., 2016).

Former work on a harbor seal stranding event using HTS

In our previous work, we used harbor seals and meta-transcriptomics to identify potential neurotropic bacteria and viruses in live stranded harbor seals that later died in a rehabilitation center (Rosales & Vega Thurber, 2015). Due to the unknown etiology of the stranding, we termed these animals involved as seals with an “unknown cause of death” or “UCD.” These seven animals had neuronal necrosis in the cortex and cerebellum, which veterinarians hypothesized was due to hypoxia, exposure to toxins, poor nutrition, or a viral infection. However, a viral etiology was thought to be the most likely culprit given the distribution and characteristics of the brain lesions, and general gross pathology of the animals.

Additionally, we used PhV-1 infected harbor seals, to benchmark our methods and to compare our analysis to animals with a described disease. Our analysis showed no evidence of a viral infection in UCD samples, but we were able to detect PhV-1 in PhV-1 infected samples. Interestingly, in our study, we found a significant presence of Burkholderia bacteria in UCD animals. Yet, necropsy reports were contradictory to this finding suggesting that either Burkholderia were part of a secondary or opportunistic infection or elusive in the original dissections (Rosales & Vega Thurber, 2015). Therefore, this leaves the cause of death of this cohort of UCD animals unresolved.

In this study, we further evaluated this dataset by using transcriptomic analysis to better understand the cause of death of the seven neonatal harbor seals that died from an unknown brain disease. Additionally, we aimed to use transcriptomics to characterize the gene expression of four neonatal harbor seals with a known PhV-1 brain infection. We hypothesized that animals responding to a PhV-1 infection should exhibit increases in host-virus response genes, while UCD animals would have characteristic gene repertoires of animals with a bacterial infection and/or hypoxia, exposure to toxins, or poor nutrition. To examine this, we looked at significant gene expression alterations of UCD and PhV-1 infected harbor seals.

Materials and Methods

This work was authorized by the National Marine Fisheries (NMFS) for possession of tissue samples from stranded marine mammals. This work is in compliance with the Marine Mammal Protection Act (MMPA) regulation 50 CR 216.22 and 216.37.

In this study, we aimed to use transcriptomics to identify the cause of death of harbor seals that died from an unidentified brain disease and to characterize host pathways of harbor seals during a PhV-1 infection. For transcriptome analysis, 11 harbor seal brain tissue samples were evaluated. The harbor seal brain tissues were kindly provided by the Marine Mammal Center (MMC) in Sausalito, CA, USA, where the animals expired. Brain tissues were stored at −80 °C and belonged to the cerebrum with the exception of sample UCD2, which was tissue from the cerebellum. Stranded animals were collected from 2009–2012, and necropsied from fresh carcasses soon after death (Table 1, Date of necropsy). Samples ranged in age at the time of stranding, from weaner (<1 month, n = 8) to pup (<1 year, n = 3). All UCD harbor seals had neuronal necrosis in the cortex and cerebellum. Other common disease signs described in these animals were: hepatic lipidosis (4/7), spleen hemosiderosis (5/7), and spleen extramedullary hematopoiesis (6/7). Table 1 details a summary of necropsy reports.

Table 1 Stranding information for harbor seal samples used in this study.

Sample ID	Date of stranding	Date of death	Date of necropsy	Common lesions	Age	Sex	Tissue	Weight in kilo- grams at necropsy	
UCD1	4/8/09	7/1/09	7/2/09	Neuronal necrosis in the cortex and cerebellum, and hepatic lipidosis	Weaner	M	Cerebrum back	9.6	
UCD2	4/9/09	7/26/09	7/29/09	Neuronal necrosis in the cortex and cerebellum, splenic hemosiderosis, spleen extramedullary hematopoiesis, and hepatic lipidosis	Weaner	F	Cerebellum front	11.0	
UCD3	4/11/09	4/21/09	4/22/09	Neuronal necrosis in the cortex and cerebellum, splenic hemosiderosis, and spleen extramedullary hematopoiesis	Weaner	M	Cerebrum front	11.9	
UCD4	4/17/09	7/6/09	7/6/09	Neuronal necrosis in the cortex and cerebellum, splenic hemosiderosis, spleen extramedullary hematopoiesis, and hepatic lipidosis	Weaner	F	Cerebrum front	13.0	
UCD5	4/20/09	7/12/09	7/13/09	Neuronal necrosis in the cortex and cerebellum, and spleen extramedullary hematopoiesis	Weaner	F	Cerebrum front	10.8	
UCD6	5/2/09	6/26/09	6/27/09	Neuronal necrosis in the cortex and cerebellum, splenic hemosiderosis, spleen extramedullary hematopoiesis, and hepatic lipidosis	Weaner	M	Cerebrum front	10.0	
UCD7	6/1/09	7/16/09	7/16/09	Neuronal necrosis in the cortex and cerebellum, splenic hemosiderosis, and spleen extramedullary hematopoiesis	Weaner	F	Cerebrum front	8.7	
PhV-1com3	3/14/10	5/2/10	5/3/10	Necrosis in the liver, adrenal gland, and lymph tissue	Weaner	M	Cerebrum front	7.5	
PhV-1com5	3/29/11	4/7/11	4/8/11	Hemorrhagic and congested lungs, mottled liver, congested meninges, intestinal necrosis, necrosis in the liver, and adrenal gland	Pup	M	Cerebrum front	7.5	
PhV-1com6	4/16/11	4/24/11	4/25/11	Fat atrophy, omphalophebitis, enlarged mesenteric lymph nodes, thickeded umbilicus, and necrosis in the liver and lung	Pup	M	Cerebrum front	11.0	
PhV-1com7	5/25/12	5/25/12	5/26/12	Omphalophebitis, necrotizing splentitis, hepatitis, and adrenalitis	Pup	F	Cerebrum front/back	8.3	

Transcriptome library preparation

Transcriptome libraries were prepared as previously published (Rosales & Vega Thurber, 2015). Briefly, a disposable pestle was used to homogenize ∼0.5ng of the frozen brain sample in TRIzol (Life Technologies, Carlsbad, CA, USA). The homogenate was centrifuged for 10 min at 12,000× g at 4 °C, and the supernatant transferred to a clean tube. For every 1 mL of Trizol, 0.2mL of chloroform was added to the supernatant, vortexed briefly, and centrifuged at 10,000× g for 18 min at 4 °C. The aqueous layer was then transferred to a clean tube and equal volumes of 100% ethanol were added to samples, and loaded onto an RNeasy column for extraction as recommended by the manufacturer (Qiagen, Hilden, Germany). To remove DNA, samples were exposed to 2U of Turbo DNase (Life Technologies) for 9 h at 37 °C. Harbor seal rRNA was removed using the Ribo-Zero Kit Gold (Human-Mouse-Rat) from Epicentre (WI, USA) following the manufacturer’s directions. High-quality RNA was converted to cDNA using superscript II Reverse Transcriptase (Life Technologies). Libraries were prepared for each of the 11 samples using the TruSeq paired-end cluster kit v.3 from Illumina (San Diego, CA, USA). Libraries were sequenced on two lanes of the Illumina Hi-Seq 2000 platform. Each lane had a random mixture of both harbor seal groups (UCD and PhV-1 infected animals).

Bioinformatic quality control and analysis

Using FqTrim the data was quality filtered with a minimum Phred score of 30. Sequences were trimmed and adapters and poly-A tails removed (Geo Pertea, 2015). In addition, post- FqTrim sequences were trimmed a second time, to ensure all sequence lengths were a minimum of 75 bps long. Transcriptome assembly was then conducted using a combination of transcriptome-guided and de novo methods. All quality assured sequence reads from both libraries were combined and aligned to the hypothetical Weddell seal, Leptonychotes weddellii, transcriptome (NCBI accession: PRJNA232772), using the program Bowtie2-2.2.3 (Langmead & Salzberg, 2012). Aligned sequences were then used to build a de novo harbor seal transcriptome using Trinity 2.0.6 with parameters –single, and –full-cleanup (Haas et al., 2013). Statistics for the assembly were obtained with Transrate v1.0.3 (Smith-Unna et al., 2016). The longest representative transcript for each component or subcomponent in the transcriptome assembly was selected using trinity_reps.pl (https://goo.gl/EGq7I6). To calculate the number of transcripts for each library, the 11 libraries were first individually aligned against the de novo transcriptome using trinity’s align_and_estimate_abundance.pl with options –aln_method bowtie2 – and –trinity_mode–prep_reference (Haas et al., 2013). The aligned sequences from each library were counted using the script SamFilter_by_components.pl (http://goo.gl/kkvqdK).

Differentially expressed genes (DEGs) between UCDs and PhV-1 infected seals

To normalize gene counts and determine differentially expressed genes (DEGs) between animals infected with PhV-1 and those with an unknown cause of death (UCD), the count data were analyzed using R version 3.2.2 with software packages Bioconductor 3.1 and DESeq2-1.8.2. For each gene, DESeq2 fits a log generalized linear model with a negative binomial distribution to normalize genes abundances (Love, Huber & Anders, 2014). Transcripts were significantly different if they had at least an adjusted p-value ≤ 0.05. Importantly, since in this study we compared two diseases with no true control samples, a positive log fold change was considered up-regulated in UCD samples, while if the log fold change was negative it was considered up-regulated in PhV-1 infected samples.

Gene Ontology (GO) enrichment analysis

The transcriptome assembly was then annotated with an e-value of ≤10−20 using GenesFromLocalDB.pl (http://goo.gl/4Zbbt5), a script that utilizes BLASTx to assign gene names to transcripts using the UniProt database downloaded in 2014 (Magrane & Consortium, 2011). Gene ontology (GO) was assigned to the annotated transcriptome with the script GOFromGeneAnnotation.pl (http://goo.gl/jJ4vg9). Transcript IDs with assigned GO terms were then combined with their respective DESeq p-values.

The software package ErmineJ 3.0.2 was then applied to evaluate the biological pathways associated with each differentially expressed GO term (Lee et al., 2005). The analysis was run with the options: gene score resampling (GSR, which does not require a threshold and thus evaluates all p-values), a maximum gene set of 100, a minimum gene set of 20, a maximum iteration of 200,000, and full resampling. GO terms with GO p-values ≤ 0.05 and a multifunctionality of ≤0.85 were semantically summarized and visualized with REViGO with an allowed similarity of 0.90, the most conservative setting (Supek et al., 2011).

KEGG analysis

For KegArray analysis, the harbor seal transcriptome was translated to protein reads with TransDecoder 2.0 (Haas et al., 2013). The program KAAS (KEGG Automatic Annotation Server) was used to annotate translated transcripts with BLASTx against a manually curated KEGG GENES database (Kanehisa, 2000; Moriya et al., 2007). The KAAS options used were ‘partial genome’ and ‘bi-directional best hit’ (BBH). KEGG ontology (KO) assignments with a respective DESeq padj value ≤ 0.05 were used for further analysis. KegArray was then utilized to map KO pathways and CytoKegg, (a Cytoscape application (http://apps.cytoscape.org/apps/cytokegg)) to visualize specific KO pathways.

Spearman correlation analysis

To evaluate correlations between the significantly high fatty acid metabolic genes in UCDs and the significantly high Burkholderia transcript abundances, we conducted a Spearman correlation analysis. All UCD normalized (by DESeq2) transcript counts that fell within the fatty acid metabolism GO category by ErmineJ were used for this analysis. Also, UCD Burkholderia normalized (by DESeq2) transcript abundance values from our previous research were obtained (Rosales & Vega Thurber, 2015). A Spearman correlation analysis was then conducted with R 3.2.2 using function cor.test.

Results

Harbor seal brain transcriptome assembly

In this study, we generated 11 harbor seal brain transcriptome libraries to distinguish genes expressed in the brains of harbor seals during a PhV-1 infection and from an unknown etiology. Of note, from here on, we will refer to PhV-1 infected samples as PhV-1 comparative or PhV-1com as they were referred to as “comparative” in previous work (Rosales & Vega Thurber, 2015). From the 11 libraries, the Hi-Seq 2,000 produced a total of 546,003,190 reads of 100 bps in length. Libraries ranged from 41,767,080 to 58,031,096 sequences, with means of 47,800,849 (SEM = 1,199,797) and 52,849,311 (SEM = 1,929,808) sequences, for UCD and PhV-1com samples, respectively. The data showed no significant difference in the number of sequences between PhV-1com and UCD datasets (Welch Two Sample t-test, p = 0.07).

To build the harbor seal transcriptome necessary for our downstream analyses, we used a combination of a transcriptome guided approach with the Leptonychotes weddellii transcriptome (NCBI accession: PRJNA232772) and de novo methods. The 11 libraries were aligned to the Leptonychotes weddellii hypothetical transcriptome. A total of 163,769,951 sequences aligned which equated to 27.43% of the total data. These sequences were then used for the de novo construction of the harbor seal transcriptome and is available on figshare (https://dx.doi.org/10.6084/m9.figshare.3581712.v1). Next, each library was aligned to the harbor seal de novo transcriptome with alignments ranging from 17.1% to 28.05% and there were no significant differences between PhV-1com and UCD alignments (Welch Two Sample t-test, p-value = 0.2825).

UCD and PhV-1 infected seals show distinct gene expression profiles

The transcriptome guided and de novo approach resulted in a harbor seal brain transcriptome of 32,856 transcripts. Next, the longest representative read was selected for each component and 29,512 transcripts remained with an average length of 269 bps. The maximum transcript length was 54,385 bps, with a minimum length of 224 bps (Figs. S1A and S1B). Using BLASTx, 25,840 (∼87.5%) transcripts had significant similarity to proteins in the UniProt database. A total of 1,962 differentially expressed genes (DEG) were identified as measured by a padj of ≤0.05 (the data is available on figshare https://dx.doi.org/10.6084/m9.figshare.3767307.v1 and https://dx.doi.org/10.6084/m9.figshare.3766986.v1). Datasets appear to have distinct gene expression profiles, with UCD samples exhibiting tighter clustering than PhV-1com samples (Fig. 1).

Figure 1 Batch effects on transcripts from the brain tissue of harbor seal samples.

Principal Coordinate Analysis (PCA) of all annotated transcripts in both PhV-1com and UCD harbor seals.

Functional annotation of differently expressed genes distinguishes UCD from PhV-1 infected harbor seals

We explored enriched gene categories in the data by performing a gene ontology (GO) analysis. In our pipeline, we identified 19,788 GO terms in the harbor seal transcriptome. After filtering based on GO term p-values and multifunctionality (values generated by ErmineJ), 32 GO terms remained and from these terms the four most significantly enriched were: (1) “antigen processing and presentation” (p-value = 1.00e−12), (2) “defense response to virus” (p-value = 1.00e−12), (3) “response to virus” (p-value = 1.00e−12), and (4) “innate immune response-activating signal transduction” (p-value = 1.00e−05). After, GO terms were semantically summarized and the categories with the most significant GO terms were: (1) “antigen processing and presentation” followed by (2) “response to amino acids,” (3) “DNA packaging,” and (4) “mononuclear cell proliferation.” The least significantly enriched GO term categories were “phagocytosis” and “fatty acid metabolism” (Fig. 2) Genes that were significantly differentially expressed genes (DEGs; padj ≤ 0.05) and found in significantly GO enrichment analysis resulted in 112 significant genes that clustered with their respective group (Fig. 3). The majority of transcripts (85.7%) in this analysis were up-regulated in the PhV-1com samples. Transcripts that belonged to the fatty acid metabolism GO category showed a higher gene expression in UCD samples (Fig. 3). Also, of particular interest, GO categories for “defense response to virus” and “response to virus” were up-regulated in PhV-1com and not UCD animals (Fig. 3). In addition, three out of four PhV-1com samples showed gene enrichment for bacterial infection, but a bacterial host response was not apparent in UCDs (Fig. S2).

Figure 2 Semantically summarized GO terms.

Tree map summary of 32 significantly enriched GO terms (p-value ≤ 0.05, and multifunctionality ≤ 0.85) of both UCD and PhV-1 samples. The blocks are clustered by related terms and the size of the boxes are based on log10 transformed p-values from GO enrichment analysis. Larger boxes represent more significant p-values.

Figure 3 Significant differentially expressed genes (padj ≤ 0.05) within GO categories that were significantly enriched in the harbor seal transcriptome (GO p-value ≤ 0.05 and multifunctionality of ≤0.85).

Heatmap of normalized gene counts expressed in rlog transformation (row z-score) from PhV1com and UCD harbor seals. Scatter plot of log2 fold change between PhV1 and UCD. The respective DEG padj values for each gene are represented by circles, with smaller circles denoting smaller padj values. Category: purple = GO term: fatty acid metabolic process, orange = GO terms: defense response to virus, and response to virus, grey = the other 29 GO terms that were significantly enriched in the harbor seal transcriptome.

KEGG analysis reveals host responses to phocine herpesvirus-1 infection

To further evaluate functional pathways found in UCD and PhV-1com disease states, we annotated the translated harbor seal transcriptome with the KEGG Automatic Annotation Server (KAAS). KAAS identified a total of 15,586 KOs from the whole transcriptome assembly and from these we extracted the 1,464 DEGs. Using KegArray, it was found that the five most significant abundant KO pathways were for: Metabolic Pathways (107 members), PI3K-Akt Signaling Pathway (43 members), Pathways in Cancer (39 members), Human T-Lymphotropic virus-1 Infection (36 members), and Herpes Simplex Infection (36 members). Given that PhV-1com samples had previously been shown to have a herpesvirus infection (e.g., PhV-1), we focused on the herpes simplex virus KEGG PATHWAY map and looked at genes up-regulated in PhV-1com harbor seals. All 36 KO terms were up-regulated in PhV-1 infected samples and partially mapped to the human herpes simplex virus-1 (HSV-1) pathway (Fig. 4).

Figure 4 KEGG pathway involved in human herpes-simplex-1 (HSV-1) showing similarities in host gene responses upon a PhV-1 infection.

Highlighted gray boxes represent terms that were significantly enriched DEGs in PhV-1 infected seals (DESeq2 padj ≤ 0.05).

Correlations of Burkholderia and UCD fatty acid genes

We further evaluated transcripts assigned to fatty acid metabolism by GO enrichment analysis. Transcripts that were significantly up-regulated in the fatty acid metabolism category (padj ≤ 0.05) in UCD animals were compared to KAAS annotation (Table 2). The transcript annotations were similar using both the UniProt database and the KEGG GENES database (Table 2). In addition, since UCD animals showed significant expression of fatty acids metabolism and our earlier study showed significant levels of Burkholderia RNA we looked for a correlation between these two factors (Rosales & Vega Thurber, 2015). A Spearman correlation of the data yielded a significant correlation of fatty acid metabolism genes and Burkholderia transcript abundance across the samples (rs = 0.809 and a p-value = 0.004).

Table 2 Transcripts in UCD samples involved in fatty acid metabolism.

Fatty acid metabolism transcripts that were significantly up-regulated (DEGs padj ≤ 0.05) in UCD harbor seals and annotated using UniProt, GO terms, KEGG GENES and KO pathways.

Gene ID	UniProt annotation	GO category	KEGG annotation	KO pathway	Fold change	Padj	
TR11985_c0	Elongation of very long chain fatty acids protein	Fatty acid metabolic process	Elongation of very long chain fatty acids protein 5	Fatty acid metabolism, biosynthesis of unsaturated fatty acids, and fatty acid elongation	0.697	0.001	
TR13138_c0	Fatty acid 2-hydroxylase	Fatty acid metabolic process and fatty acid biosynthesis	4-hydroxysphinganine ceramide fatty acyl 2-hydroxylase	NA	0.99	0.003	
TR5359_c0	Fatty acid desaturase 2	Fatty acid metabolic process and fatty acid biosynthesis	Fatty acid desaturase 2	PPAR signaling pathway, fatty acid metabolism, biosynthesis of unsaturated fatty acid, and alpha-Linolenic acid metabolism	0.49	0.011	
TR15982_c0	Long-chain specific acyl-CoA dehydrogenasemitochondrial	Fatty acid metabolic process	Long-chain-acyl-CoA dehydrogenase	NA	0.469	0.007	
TR7794_c0	Long-chain-fatty-acid–CoA ligase 1	Fatty acid metabolic process	Long-chain acyl-CoA synthetase	Fatty acid biosynthesis, fatty acid degradation, fatty acid metabolism, PPAR signaling pathway, Peroxisome, and adipocytokine signaling pathway	0.424	0.036	
TR9787_c0	Medium-chain specific acyl-CoA dehydrogenasemitochondrial	Fatty acid metabolic process	Acyl-CoA dehydrogenase	Fatty acid metabolism, PPAR signaling pathway, Carbon metabolism, beta-Alanine metabolism , valine, leucine isoleucine degradation, Fatty acid degradation, and propanoate metabolism	0.831	2.37E–06	
TR283_c0	Stearoyl-CoA desaturase variant (Fragment)	Fatty acid metabolic process and fatty acid biosynthesis	Stearoyl-CoA desaturase	AMPK signaling pathway, fatty acid metabolism, PPAR signaling pathway, biosynthesis of unsaturated fatty acids, and longevity regulating pathway—worm	1.354	1.28E–13	
TR1355_c0	Sterol-C4-methyl oxidase-like protein (Fragment)	Fatty acid metabolic process	Methylsterol monooxygenase	Steroid biosynthesis	0.856	0.003	
TR10658_c1	Sterol-C5-desaturase-like protein (Fragment)	Fatty acid metabolic process and fatty acid biosynthesis	Delta7-sterol 5-desaturase	Steroid biosynthesis	0.957	9.69E–05	
Notes.

PPAR peroxisome proliferator-activated receptors.

AMPK adenosine monophosphateactivated protein kinase

Discussion

In marine mammals, transcriptomics has never been used to comprehend the cause of an unknown disease and rarely has it been used to characterize the global gene expression of known marine mammal stressors (Mancia et al., 2014; Niimi et al., 2014; Khudyakov et al., 2015a; Khudyakov et al., 2015b; Fabrizius et al., 2016). Here, we used transcriptomics to compare gene expression patterns to known and unknown disease states of stranded harbor seals. We infer the cause of a brain disease in seven young harbor seals and characterize host pathways involved during a PhV-1 infection in the brains of four young harbor seals. Gene expression of harbor seal brains with an unknown cause of death (UCD)

As stated earlier, the initial hypothesis for the root cause of death of UCD harbor seals was a viral infection. However, exposure to toxins, nutrient depletion, and hypoxia were also candidates for the death of these animals. In our former work, we showed that a viral infection was unlikely the cause of mortality in UCD harbor seals (Rosales & Vega Thurber, 2015). We further confirmed this by demonstrating that GO categories for “defense response to virus” and “response to virus” were not expressed in UCD animals (Fig. 3). At the same time, we validated that UCD animals had a similar gene response at the time of death, thus supporting the notion that these harbor seals died from the same disease (Fig. 1).

UCD harbor seal gene response to bacteria

In our previous work on this data set, we also found that there was a significant abundance of Burkholderia transcripts in UCD animals and our new results indicate that these same animals exhibit high fatty acid metabolic process gene expression (Fig. 3). In this study, we found a significant correlation between Burkholderia and fatty acid genes. It is possible that fatty acid metabolism is triggered by and/or provides an environment that promotes the growth of Burkholderia, but substantial research needs to be conducted to confirm this correlation. To our knowledge, there is no documentation of Burkholderia increasing due to high fatty acid production, but there is evidence that Burkholderia can grow competitively in humans during metabolically stressful situations (Schwab et al., 2014).

In addition, in this study, we found that significantly expressed DEGs for “response to bacteria” were up-regulated in the majority of PhV-1 infected samples (three out of four) and not upregulated in UCDs. Since UCD samples had a significant abundance of Burkholderia it was expected that UCD animals would have an upregulated gene expression to “response to bacteria” (Fig. S2). However, in our previous study, we noted that the microbiome was significantly less abundant in UCD animals when compared to PhV-1 infected animals. Thus it is likely that the low abundance of bacteria in UCD animals compared to PhV-1 infected animals drives this gene expression pattern.

Fatty acid metabolism associated with harbor seal strandings

To reiterate, in our GO summary analysis there was no indication of a viral infection, but we did find “fatty acid metabolism” genes enriched in UCD animals. In fact, the DEG analysis demonstrated that this GO group was the most significantly up-regulated category in UCD animals (Fig. 3). Using KEGG analysis, we further substantiated the involvement of these genes in fatty acid metabolism (or related pathways involved in lipid and fat metabolism e.g., steroid biosynthesis) (Table 2). Fatty acid metabolism genes are important for fundamental cellular functions such as those involved in the formation of phospholipids and glycolipids, as well as in the energetics for the cell cycle, like cell proliferation, differentiation, and energy storage.

In marine mammals, esterified fatty-acids (NEFA) can be used as a proxy for nutritional health (Trites & Donnelly, 2003). For example, if gray seal pups fast for over a month they show elevated NEFAs and reduced glucose (Rea et al., 1998). Although, given that UCD samples were in a rehabilitation center and had normal weight measurements (Table 1), UCD harbor seals do not appear to have died from starvation. In mammals, fatty acids are mostly acquired through dietary means except in the liver and adipose tissue where fatty acid pathways are utilized (Kuhajda, 2000). Thus, high fatty acid gene activity, in regions other than the liver or adipose tissues, can be symptomatic of metabolic diseases other than starvation. As an example, cells with up-regulated fatty acid synthase (FAS) can be a sign of tumorigenesis (Kuhajda, 2000).

A possible mode of death for UCDs is that these animal were unable to adequately take in nutrients since a lack of adequate dietary intake of some fatty acids can lead to an increase in fatty acid metabolism in the brain (Innis, 2008). To illustrate, if an animal has an insufficient intake of ω-3 fatty acids, then the brain increases in ω-6 fatty acid content. In a developing brain, this increase in ω-6 fatty acids can lead to problems with neurogenesis, neurotransmitter metabolism, and altered learning and visual function. Metabolic disorders are commonly reported in cetaceans with hepatic lipidosis or fatty liver disease (Jaber et al., 2004). Interestingly, UCD necropsies reported that four animals had hepatic lipidosis, which is a disease attributed to toxins, starvation, or nutrient deprivation in weaning animals (Jaber et al., 2004).

Fatty acids, specifically, can be used to detect chemical or toxic stress in marine organisms (Filimonova et al., 2016). Since, these animals did not appear to be starved, this suggest that nutrient depletion or toxin exposure may have been involved in the die-off of UCDs because (1) these were neonatal harbor seals (with a developing brain), (2) the coincident description of the necropsy reports, and (3) the fatty acid metabolic shifts in the brains of these animals. Other common lesions found in UCDs were spleen hemosiderosis and spleen extramedullary hematopoiesis. These syndromes have been associated with other metabolic diseases, but we are unsure if they are directly related to high fatty acid gene expression in the brains of these neonatal seals.

As mentioned, fatty-acid markers have been used to detect stress responses in marine organisms (Trites & Donnelly, 2003; Filimonova et al., 2016). The transcripts detected in this study have the potential to be used as biomarkers for stranded animals with an elusive etiology or marine mammals that died from necrosis of the brain tissue. Gathering such information may aid in better understanding this mysterious disease and help to properly diagnose other animals.

Gene expression of harbor seal brains infected with phocine herpesvirus-1

The gene response of harbor seals infected with PhV-1 is mostly unknown, but our data now illuminates some understanding of this interaction. The HSV-1 KEGG Pathway (Fig. 4) shows evidence that PhV-1 promotes some host gene responses similar to other viruses in the subfamily Alphaherpesvirinae. Although, it appears that there are still many pathways that differ between HSV-1 and PhV-1. However, it is likely that PhV-1 host response may better parallel other viruses from its genus Varicellovirus, like bovine herpesvirus-1 (BHV-1). In BhV-1, the host immune system has been shown to respond in three stages: early cytokines, late cytokines, and cellular immunity or adaptive immunity (reviewed in Babiuk, Van Drunen Littel-van den Hurk & Tikoo, 1996). Although our data is non-temporal, the summarized enriched GO analysis illustrates evidence for aspects of each of these three predefined temporal stages (Fig. 2). For example, “response to amino acid” (Fig. 2 pink blocks) provides evidence of the early cytokine immune responses in stage 1. At the same time our DEG analysis shows that Toll-like receptors (TLR) are significantly expressed in these animals (p-value < 0.001); thus we speculate that TLR3 and TLR7 may be involved in the detection of PhV-1 in harbor seal brain cells (Fig. 3). TLR7 is part of a TLR group that can detect viral Pathogen- associated molecular patterns (PAMPs) within endosomes and lysosomes (Heil et al., 2004) and TLR-3 is known to activate an antiviral state within an infected cel1 (Tabeta et al., 2004). Thus, we reason that TLR3 and TLR7 ultimately lead to the induction of a nonspecific positive regulation inflammatory response seen in these animals (Fig. 2, pink blocks “positive regulation of inflammatory response”).

Furthermore, cell chemotaxis, leukocyte chemotaxis, and interleukin-6 are also important early cytokine stage responses found in our data (Fig. 2, pink blocks) (Babiuk, Van Drunen Littel-van Den Hurk & Tikoo, 1996). Cell chemotaxis and leukocyte chemotaxis are needed for recruitment of cells and could be responsible for attracting cells to the site of a PhV-1 infection, while interleukin-6 promotes macrophage differentiation. Differentiated macrophages can then secrete cytokines, like tumor necrosis factor (TNF) (Fig. 2. pink blocks). The early and late stage cytokine activity is depicted in the summarized GO category “mononuclear cell proliferation” (Fig. 2 yellow blocks). Once at the site of infection leukocytes are likely to proliferate, while late stage cytokines can cause proliferation of mononuclear cells, such as T-cells, B cells, and Natural Killer cells (NK cells).

The most pronounced category in this data is the last stage or cellular immunity (Fig. 2 orange blocks). Antigen processing and presentation is an important step in developing cellular immunity, which occurs when an antigen, like PhV-1, is processed into proteolytic peptides and loaded onto MHC class 1 or II molecules on a cell. We found that “antigen processing and presentation” is a highly enriched GO category (p-value <0.001) and that transcripts for MHC I and II are highly expressed in PhV-1 infected samples (Fig. 3, padj < 0.0001), demonstrating that the immune system of these young harbor seals was able to develop an adaptive immune response to PhV-1. Furthermore, the “phagocytosis” category data, suggest that a cellular mechanism to clear PhV-1 infected cells in harbor seals brains might be phagocytosis, as it was an enriched GO category (Fig. 2, teal blocks). However, we cannot refute the possibility that phagocytosis may be a route for viral entry into the cell. Recently in equine herpesvirus-1 (EHV-1), from the genus Varicellovirus, there was an indication of a phagocytic mechanism for EHV-1 to enter some cells (Laval et al., 2016). Alternatively, or in conjunction, PhV-1 may have appropriated the host exocytosis pathway to egress from the cell, as has been noted in other alphaherpesviruses (Fig. 2, teal block “regulation of exocytosis”) (Hogue et al., 2014).

DNA packaging during a PhV-1 infection

Another category enriched in our GO analysis was “DNA packaging” (Fig. 2, green block), which occur when a chromatin structure is formed from histones to create nucleosomes (Felsenfeld, 1978). Here, we predict that PhV-1 hijacked the host DNA packaging pathway. Presently, there is controversy about the role of DNA packaging during herpesvirus infections. At least three different states of DNA packaging occur during a herpesvirus infection. Within the viral particle the double stranded genome is not packaged, but in the latent state of the virus, it associates with cellular nucleosomes forming a cellular chromatin-like structure (Lee, Raja & Knipe, 2016). The controversy arises from the lytic or replication cycle. Studies show varying degrees of chromatin with herpesvirus DNA and these variations in chromatin may be associated with viral transcription (Herrera & Triezenberg, 2004; Lacasse & Schang, 2012; Lee, Raja & Knipe, 2016). We suspect that PhV-1 was either entering the latent phase and/or that chromatin formation was occurring because of active viral transcription.

Of interest within the “DNA packaging” category the most significantly up-regulated histone is H3.2 like protein (Fig. 3, p-value > 0.0001), a variant of histone H3. Some variants of H3, like H3.3, have been shown to be important during herpesvirus transcription, the role of H3.2 in herpesvirus is more elusive (Placek et al., 2009). Although, the role of H3.2 in the latent phase cannot be disregarded since H3 has been associated with both the latent and lytic phases (Kubat et al., 2004; Wang et al., 2005; Kutluay & Triezenberg, 2009). Our results suggest that DNA packaging is important for PhV-1, but the exact role of DNA packaging in PhV-1 requires further research.

Caveats and considerations

Marine mammal diseases can be difficult to diagnose given their protected status and the challenge to gather conventional control samples for studies. For this research, we used two disease states, one known and one unknown. Our study shows that this method can yield valuable insight into host responses to infection, but we recognize the limitations to this approach. For instance, we were limited to evaluating up-regulated genes and consequently, we did not evaluate any down-regulated genes that may have been meaningful for understanding the diseases.

In addition, it is probable that there are shared genes or pathways in both diseases, and this commonality would not have been apparent in our DEG analysis between the two groups. As an example, it is known that host fatty acids are also up-regulated during viral infections; thus fatty acid DEGs in the UCD animals may actually have been even more numerous had we compared UCD animals with a different group of animals that did not have a viral infection (Jackel-Cram, Babiuk & Liu, 2007; Heaton et al., 2010; Spencer et al., 2011).

Finally, there is the potential that genes identified as up-regulated in one disease state are actually a result of down-regulated genes in the other disease state. However, since we knew that PhV-1 was an infectious agent in one cohort of animals and since we attained necropsy reports with probable causes of UCD disease, we were able to confidently tease apart our results with this information. Optimistically, with the increased use of HTS methods, we expect that more transcriptome studies on marine mammals will become available and this may help diminish these caveats.

In addition, it is apparent that the sequence alignment rates in this study were low and this is likely because alignments were conducted using the Weddell seal transcriptome and not the genome. Using the Weddell seal genome, a greater portion of the data aligned (76.51%). Our results are in compliance with previous research where alignments to the transcriptome are lower than alignments to the genome of an organism (Conesa et al., 2016). In spite of this, it is still a valid approach to use the transcriptome with the caveat that novel genes are not likely to be identified (Conesa et al., 2016). In this study, we did not use a genome-guided approached since this method resulted in up to 1,323,851 transcripts, which is overly abundant. In addition, in the genome-guided approach, only 4.9% of the data represented ORFs and the N50 score was 1,136. However, the transcriptome guided method resulted in 32,856 transcripts with 65.9% of transcripts accounting for ORFs and an N50 of 1,994 (Table S1).

Moving forward, we like to acknowledge that we used a small sample size, especially for PhV-1 infected animals (N = 4). Including a larger sample size could elucidate other trends in these diseases, such as the effects of gender (if any) or make correlations more apparent. In addition, it is important to note that PhV-1 typically infects the adrenal glands of seals and the infection does not always reach the brain (Gulland et al., 1997; Goldstein et al., 2005). Since viruses infect organs differently, PhV-1 may not replicate in the same manner in the brain as it does the adrenal glands. Future studies may focus on comparing transcriptomes from the brain, adrenal glands, and other PhV-1 affected organs to determine any differences between the host organs and virus interactions.

Conclusion

This is the first study to evaluate transcriptomes to better understand virus-host interactions and brain tissue responses to an unknown disease in marine mammals. In samples with a PhV-1 brain infection, we identified pathways involved in innate and adaptive immunity, as well as DNA packaging transcripts. We now have a better understanding of PhV-1 gene expression in brain tissue of pinnipeds, which may lead to improved management and treatment of PhV-1 infections. However, more work including time series data is needed to comprehend the mechanism and progression of this disease. In addition, with this analysis, we were able to further confirm our results, from our previous work, that UCD animals did not die from a viral infection. Instead, we found that fatty acid metabolic genes were highly up-regulated in UCD animals. It is unknown what may have caused a manifestation of fatty acid metabolism dysregulation in the brains of UCD harbor seals, but it is probable that it may have been linked to exposure to toxins or nutrient depletion.

Supplemental Information

Figure S1 Length and counts of the harbor seal brain transcriptome.

(A) The length and count distribution of the harbor seal brain transcriptome with transcript lengths < 8,500 and (B) the length and count distribution of the harbor seal brain transcriptome with transcript lengths > 8,500.

Click here for additional data file.

Figure S2 Significant differentially expressed genes (DEGs, padj < 0.05) that fell within GO category “defense response to bacteria.”

Heatmap hierarchical clustering of normalized gene counts expressed in rlog transformation (row z-score) from PhV1com and UCD harbor seals.

Click here for additional data file.

Table S1 Transcriptome assemblies conducted using quality controlled sequence reads from both libraries (UCD and PhV-1com)

Click here for additional data file.

We thank the Marine Mammal Center in Sausalito, CA, USA, for providing harbor seal tissue samples and necropsy reports. We also like to thank Dr. Eli Meyer at Oregon State University for his mentorship and assistance with transcriptome analysis.

Additional Information and Declarations

Competing Interests

Author Contributions

Animal Ethics

Ethics

DNA Deposition

Data Availability

The authors declare there are no competing interests.

Stephanie M. Rosales conceived and designed the experiments, performed the experiments, analyzed the data, contributed reagents/materials/analysis tools, wrote the paper, prepared figures and/or tables.

Rebecca L. Vega Thurber contributed reagents/materials/analysis tools, reviewed drafts of the paper.

The following information was supplied relating to ethical approvals (i.e., approving body and any reference numbers):

This work was authorized by the National Marine Fisheries (NMFS) for possession of tissue samples from stranded marine mammals. This work is in compliance with the Marine Mammal Protection Act (MMPA) regulation 50 CR 216.22 and 216.37.

The following information was supplied relating to ethical approvals (i.e., approving body and any reference numbers):

This work was authorized by the National Marine Fisheries (NMFS) for possession of tissue samples from stranded marine mammals. This work is in compliance with the Marine Mammal Protection Act (MMPA) regulation 50 CR 216.22 and 216.37.

The following information was supplied regarding the deposition of DNA sequences:

Rosales, Stephanie (2016): Brain transcriptomes of harbor seals demonstrate gene expression patterns of animals undergoing a metabolic disease and a viral infection. figshare.

https://dx.doi.org/10.6084/m9.figshare.3581712.v1

The following information was supplied regarding data availability:

The raw sequence data supporting the conclusions of this article is available in the European Nucleotide Archive repository, using accession number: PRJEB11686 (http://www.ebi.ac.uk/ena/data/view/PRJEB11686)

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
