# Peer review of "Brain transcriptomes of harbor seals demonstrate gene expression patterns of animals undergoing a metabolic disease and a viral infection"

_PeerJ, doi:10.7717/peerj.2819_

## Round 0.1 · original submission · Major Revisions

Overall I enjoyed reading this manuscript. It was mostly well written and put into context.

Please read the comments of both reviewers carefully they did an excellent job of providing suggestions to improve the manuscript. In particular reviewer 1 has suggested some improvements to the statistical analysis that could be performed. Reviewer 2 also highlights that the rationale for the project can get a bit lost in the introduction.

Please consider all the reviewers' comments and revise the manuscript accordingly, I look forward to the revision.

·

Basic reporting

Overall this was a very interesting study that presents a valuable gene sequence resource for a commonly studied phocid species from a tissue that is not easily available to marine mammal researchers. The results show interesting changes in gene expression consistent with tissue responses to viral infection in PhV-1 infected seals and changes in metabolism in seals that died of unknown causes.

I found several grammatical and terminology errors that I point out in the attached review document and suggest thorough revisions to most of the figure and table legends.

The raw sequenced reads should also be submitted to NCBI's SRA. The transcriptome assembly should be uploaded to a public site such as TSA or made available through a permanent host (figshare, for instance). The complete lists of DEGs (list for PhV-1-upregulated, list for UCD-upregulated, for instance) with annotation (not just GO categories) should be made available on figshare or added as supplementary files.

Experimental design

I think the experimental design was robust, although I suggest some changes to statistical analyses and additional information for several of the methods.

Validity of the findings

I am concerned about the low mapping rates of reads to transcriptome assembly. I discuss this in more detail in the attached review document.

Additional comments

See review document.

Reviewer 2 ·

Basic reporting

The authors have compared the brain transcriptomes of 4 stranded seals with phocine herpes virus and 7 stranded seals with an unknown cause of death. They investigate differences in gene expression and function between the two disease states and attempt to identify the cause of death in seals for which it is unknown and better characterize the host immune response against the phocine herpes virus. I would be supportive of publication in PeerJ if the authors are able to address a number of issues mostly related to clarity and presentation of the results.

Abstract: I would suggest rewriting Lines 17-21 of the abstract in similar way to the first section of the introduction to improve flow and precision. Additionally, do you mean that over half of marine mammal deaths go undescribed or that over half of stranded marine mammal deaths go undescribed?

Line 39: I would suggest using the word cause rather than culprit here and elsewhere in the manuscript.

Line 42: Suggest using the word unknown rather than enigmatic.

Line 52: ‘of an illnesses’ should be ‘of an illness’ or ‘of illnesses’

Line 62: Khudyakov et al. 2015, BMC Genomics describe the transcriptome of a northern elephant seal and investigate stress related changes in gene expression. This probably deserves a mention here. Furthermore, although not transcriptomes, it may be worth mentioning studies that use qPCR to detect nutritional/immune stress responses in seal species for which there are a few.

Line 63: Suggest removing ‘For instance’.

Line 80: Should be ‘stranding’ and not ‘standing.’

Lines 100-101: This sentence seems out of place.

Line 116: Should be ‘details a summary of the necropsy reports’ and not ‘these’ as these refers to just the UCD samples previously described.

Line 137: I think this should be hypothetical transcriptome and not hypothetical transcript. Same goes for Line 196.

Line 138: Should be ‘Weddell’ and not ‘wedell’.

Line 138: Should be ‘NCBI accession number’.

Line 140: Should this be --full-cleanup and not -fullclean?

Line 146: Use a url shortener?

Line 187: I would be inclined to clarify here that it is the brain transcriptome of the harbor seal.

Line 204: There should be a title here to describe this section of the results.

Line 205: Supplemental figure 1 A and B both need to be described in the legend.

Line 208: Suggest changing to ‘Datasets appear to have distinct gene expression profiles, although the UCD…’

Figures 2,3 & 4: I think that these results could be incorporated into one main figure. My understanding is that Figure 2 describes the significantly enriched GO terms of the composite transcriptome where larger boxes correspond to lower p-values. I would argue that presenting this result in such a way is not interesting in the context of the study. Instead I would incorporate this information into Figure 3. As it is, the heat map shows how 84.9% of transcripts were up regulated in PhV-1 samples but contains no interesting information regarding functional categories – the reader has to go to Supp. Table 1 to assign function to each of the arbitrary transcript names listed on the y-axis. Furthermore, Fig 4 contains the same information but cannot be compared. Instead, I would cluster and label each transcript by their GO terms shown in Figure 2 or by whichever hierarchical level of annotation makes sense. It would then be nice to include a horizontal bar plot next to each category to show the relative significance levels of each functional category (with the p-values used in Fig 2 for example). After all, the interesting result of the study is how differential expression changes between both seal groups by functional category. If I’m not mistaken I think this would also mean that Figure 4 A, B and Supp Fig 2 would not be necessary.

Line 225: Fig. 4 A and B should be switched round to comply with the order they appear in the text.

Line 272: Should be Fig. 4B.

Line 279: Remove comma after although.

Line 283: The PhV-1com samples have been referred to as the comparative samples. This is not in line with the rest of the manuscript.

General: Have the transcriptome data been deposited somewhere?

Experimental design

General: Overall, the manuscript is quite well written however I found the motivation and aims of the study difficult to establish. Whilst the comparison between PhV1 and UCD seal transcriptomes is well balanced in the results and a discussion section, the emphasis in the abstract is on the UCD story whilst in the introduction it is on the PhV1 story. The manuscript could be improved by bringing these two stories together more coherently.

Lines 96-101: This section should be a new paragraph and should be improved for clarity. Whilst the study design has already been introduced when describing the previous meta-transcriptomics study, I think it would benefit the reader if you really spelled out that you did in the current study again. I would begin with something similar to lines 107-109 which are much clearer and then follow with your hypotheses. I would also be cautious when stating you will identify the cause of death of UCD seals.

Line 180-185: The regression analysis is unclear and needs to be put into context. Why did you look at transcripts that fell within the fatty acid metabolism GO category? ‘The average of the fatty acid normalized transcripts’ – average what? Whilst this is explained in more detail in the results, a lot of this should be included in the methods.

Line 213: Filtering for what? This seems like a vast reduction in the number of GO terms. Please comment.

Validity of the findings

Figure 1: Figure shows a PCA plot of ‘the most variable 2,500 transcripts’. How were these determined and why not just use the 1,962 genes that were differentially expressed?

Line 199: Could you clarify what these alignment rates are describing? Are they describing the percentage of the harbor seal transcripts that map to the Weddell seal transcriptome or the proportion of the Weddell seal transcriptome that was mapped to by the transcripts. If the former is true, these alignment rates seem quite low for quite a closely related species. Do the authors have an explanation for this?

General: It would be worth noting the small sample size and its limitations in the discussion.

General: The authors don’t discuss their findings in the context of biomarkers for disease however I think this deserves a mention in the discussion.

---

## Round 0.2 · Minor Revisions

Please address the remaining minor changes requested by the reviewer.

Reviewer 2 ·

Basic reporting

The revised ms is much improved in terms of clarity and reporting. Both the abstract and introduction now read well and introduce the study in a balanced and clear manner. The authors have also responded thoroughly to comments regarding the presentation of results.

Experimental design

The authors have responded to comments regarding experimental design.

Validity of the findings

The authors have addressed concerns on the validity of the findings and have elaborated the discussion to include these.

Additional comments

This revised ms is well written and addresses all of my concerns from the first round of review. In particular the introduction and abstract are much clearer. I was pleased to see that my comments regarding the figures were addressed and as a result I feel the results section is much improved. I am glad to see that the discussion section has been elaborated to both address major concerns and broaden the scope of the ms.

The only remaining points I have are relatively minor. Firstly, reducing of GO annotations to 'slimmed' down versions in Fig 2. might make the functions more intuitive for the reader. Secondly, the grammar and spelling should be checked throughout.

---

## Round 0.3 · accepted · Accept

The manuscript has been improved greatly and was a pleasure to edit and read.